# Referring Image Segmentation via Joint Mask Contextual Embedding Learning and Progressive Alignment Network

**Ziling Huang**
The University of Tokyo
huangziling@nii.ac.jp

**Shin'ichi Satoh**
National Institute of Informatics, Japan
satoh@nii.ac.jp

## Abstract

Referring image segmentation is a task that aims to predict pixel-wise masks corresponding to objects in an image described by natural language expressions. Previous methods for referring image segmentation employ a cascade framework to break down complex problems into multiple stages. However, its limitations are also apparent: existing methods within the cascade framework may encounter challenges in both maintaining a strong focus on the most relevant information during specific stages of the referring image segmentation process and rectifying errors propagated from early stages, which can ultimately result in sub-optimal performance. To address these limitations, we propose the Joint Mask Contextual Embedding Learning Network (JMCELN). JMCELN is designed to enhance the Cascade Framework by incorporating a Learnable Contextual Embedding and a Progressive Alignment Network (PAN). The Learnable Contextual Embedding module dynamically stores and utilizes reasoning information based on the current mask prediction results, enabling the network to adaptively capture and refine pertinent information for improved mask prediction accuracy. Furthermore, the Progressive Alignment Network (PAN) is introduced as an integral part of JMCELN. PAN leverages the output from the previous layer as a filter for the current output, effectively reducing inconsistencies between predictions from different stages. By iteratively aligning the predictions, PAN guides the Learnable Contextual Embedding to incorporate more discriminative information for reasoning, leading to enhanced prediction quality and a reduction in error propagation. With these methods, we achieved state-of-the-art results on three commonly used benchmarks, especially in more intricate datasets.

## 1 Introduction

Referring image segmentation is a task that aims to predict a pixel-wise mask for objects referred to in

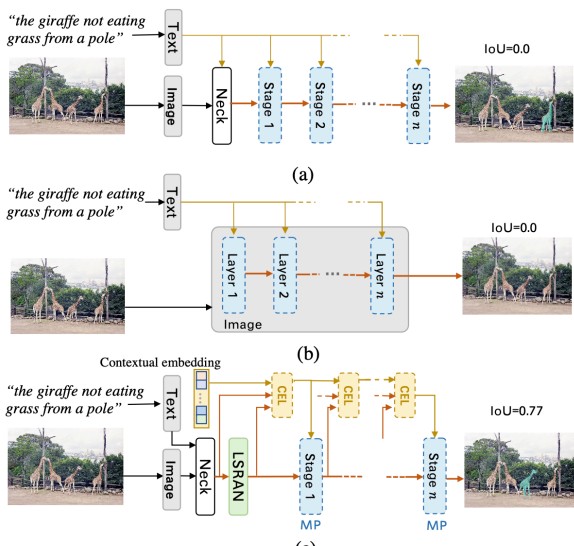

Figure 1: Comparison of our work and previous work. (a) Representative late fusion cascade networks (i.e. CGAN(Luo et al., 2020a) and CRIS(CRI, 2022)). The results shown here are from CRIS. (b) Cascade networks with early fusion (i.e. LAVT(Yang et al., 2022) and EFN(Feng et al., 2021)). The results are from LAVT. (c) The proposed network. CEL: Contextual Embedding Learning Learning. MP: Mask Predicting. The Yellow Arrows show the Contextual Embedding Learning, while The Brown Arrows show the Mask Predicting.

a natural language expression. It is one of the most basic tasks in human-robot interaction(Dautenhahn, 2007; Goodrich et al., 2008). In particular, it helps robots understand commands and localize corresponding objects. Unlike semantic/instance segmentation(Strudel et al., 2021; Chen et al., 2018), which focuses on the image only, referring image segmentation involves two different modalities: image and text. Thus, in referring image segmentation, how to achieve a good alignment between the image and the most beneficial information in text is essential for the mask prediction.

The prevailing methods in the field of referring image segmentation rely on the cascade framework, where static-text are merged with feature maps produced by the current stage to generate the output

of the subsequent stage. As illustrated in Fig. 1(a) and (b), the late and early fusion frameworks are the two commonly used cascade frameworks. The late fusion framework, as adopted in methods like CGAN and CRIS(Luo et al., 2020a; CRI, 2022), extracts text and image features separately, and then integrates them to form multi-modality features for reasoning. On the other hand, the early fusion framework, employed in methods such as LAVT and EFN(Yang et al., 2022; Feng et al., 2021), extracts language features and conducts the entire fusion operation in the image encoding backbone. The Cascade framework is a methodology that involves breaking down complex problems into multiple stages or steps, with each step building upon the previous ones. This approach can enhance the salience or prominence of referent objects in multi-stage reasoning processes. Despite their effectiveness, its limitations are also apparent: (a) If the static text used in each stage of the cascade framework is overly complex or does not appropriately focus on the most relevant information for that particular stage, it can hinder the effectiveness of the approach. For instance, consider the expression in Fig. 1, "The giraffe not eating grass from a pole." A human observer would inspect the "giraffe" in the image and utilize additional attributes to localize the target. (b) The mistakes made in the early stages of the Cascade Framework can propagate and potentially affect the later stages. Since the reasoning process in the Cascade Framework is built upon previous stages, any errors or incorrect information introduced early on can impact subsequent stages and potentially lead to incorrect conclusions or results.

To address the aforementioned issue, we propose the Joint Mask Contextual Embedding Learning Network (JMCELN). JMCELN is designed to enhance the Cascade Framework by incorporating a Learnable Contextual Embedding and a Progressive Alignment Network (PAN). The primary objective is to improve the accuracy of mask prediction by dynamically storing and utilizing reasoning information for each subsequent step. The Learnable Contextual Embedding in JMCELN allows for the dynamic storage and utilization of reasoning information. It is updated based on the current mask prediction results, rather than relying on static natural language expressions. This adaptability enables the network to capture and refine pertinent information effectively, leading to

improved mask prediction. Additionally, we introduce the Progressive Alignment Network (PAN) as part of JMCELN. PAN operates by using the output from the previous layer as a filter for the current output. It addresses inconsistencies between predictions from different stages by reducing the response of the current output and guiding the Contextual Embedding to include more discriminative information for reasoning. This iterative alignment process enhances the quality of predictions and reduces the propagation of mistakes. By combining the Learnable Contextual Embedding and PAN, JMCELN achieves dynamic reasoning and refinement of mask predictions. This integrated approach overcomes the limitations of static text and offers a powerful solution for enhancing the salience and accuracy of referent objects within multi-stage reasoning tasks.

In summary, cascade network presents two key challenges: the repetition of static text accumulates noise throughout reasoning and misunderstanding reasoning at earlier stages propagate, causing inaccurate predictions in subsequent stages. To avoid introduce too much noise, we introduce Learnable Contextual Embedding. Learnable Contextual Embedding aims to provide each stage dynamic and the most pertinent information, while PAN combats inconsistencies between stages to avoid error propagation. Learnable Contextual Embedding and Progressive Alignment Network ensure the successful operation of our error correction mechanism within the cascade network.

To evaluate the effectiveness of the proposed methods, we conduct extensive on three commonly used benchmarks and we achieved state-of-the-art results especially in in more intricate datasets.

In conclusion, our main contributions are as follows:

- We designed a Joint Mask and Contextual Embedding Learning Network (JMCELN) in which replace static-text with Learnable Contextual Embedding for stage reasoning, which endows the network with the ability to capture pertinent information for mask predicting in every stage.

- To obtain a more accurate mask, we propose the Progressive Alignment Network (PAN). If the predictions of the two stages are inconsistent, the PAN will reduce response of the later output and then force the Contextual Embed-

ding to generate more discriminative information.

- Our methods achieved state-of-the-art performance on three commonly used datasets: Refcoco(Yu et al., 2016), Refcoco+(Yu et al., 2016) and Refcocog(Mao et al., 2016), especially in in more intricate datasets, Refcoco+(Yu et al., 2016) and Refcocog(Mao et al., 2016). Extensive ablation studies demonstrated the validity of each of the proposed components.[1].

## 2 Related Work

**Referring Image Segmentation** The task of referring image segmentation was first proposed by Hu et al. (Hu et al., 2016). All work in this area can be categorized into two types. The first type includes one-stage image and language fusing of different layer feature maps in the image encoding backbone followed by a multi-layer feature exchange for better results. For example, RMI(Liu et al., 2017) and DMN(Margffoy-Tuay et al., 2018) utilize a memory unit(Hochreiter and Schmidhuber, 1997; Lei et al., 2018) to fuse individual words with visual features. CMPC(Huang et al., 2020) first perceives all entities in the image and then suppresses unrelated ones by graph reasoning. CMSA(Ye et al., 2019) has a cross-modal self-attention module to capture long-range dependencies between image and language. BRINet(Hu et al., 2020) utilizes vision-guided linguistic attention to get useful language information and then language-guided visual attention to get the final outputs. BUSNet(Yang et al., 2021) uses a bottom-up shift that progressively locates the referent. All these methods work separately on feature maps from different layers in the image encoding backbone. Then, they fuse the feature maps produced by different layers by using a memory unit(Liu et al., 2017; Margffoy-Tuay et al., 2018; Huang et al., 2020) or multi-level feature exchange(Ye et al., 2019). Not so much work implements methods like those above anymore because a one-stage interaction between visual and language information is not enough to predict accurate pixel-wise masks. The more recent second type of method aims at achieving better language and visual alignment. In particular, CGAN(Luo et al., 2020a) has a cascade framework in which language features can fuse many times with multi-modality

features output from the previous layer. CRIS(CRI, 2022) utilizes a similar framework as CGAN(Luo et al., 2020a), while it benefits from a pre-trained CLIP(Radford et al., 2021a) model. EFN(Feng et al., 2021) and LAVT(Yang et al., 2022) show that better cross-modal alignments can be achieved through early fusion of linguistic and visual features. VLT(Ding et al., 2021) generates multiple sets of word attention weights to represent different understandings. All of these methods employ static text to fuse with multi-modality features, while the information required in every stage is different.

**Dynamic Query based Segmentation** Inspired by DETR(Carion et al., 2020), some studies have proposed to use a zero/random initial vector to hook helpful information from the image feature map in semantic/instance/panoptic segmentation. MaskFormer(Cheng et al., 2021) and Mask2Former(Cheng et al., 2022) treat segmentation as a mask classification task and use a random initial query to generate information from the image feature map for final segmentation. K-Net(Zhang et al., 2021) employs semantic kernels and instance kernels as dynamic kernels to generate helpful information stage-by-stage from the image encoding backbone (i.e. the CNN and Transformer) for the final mask prediction. Panoptic SegFormer(Li et al., 2022) proposes to use things queries and stuff queries to store different types of information. All these studies verify that learnable embedding can catch demanding information based on the task requirements. Our study is the first to introduce a learnable embedding which can be updated adaptively in accordance with the requirements of each stage for stage reasoning.

## 3 Methods

The overall framework is presented below. Then, we detail the image and text encoder, LSRAN, and JMCELN. Finally, the upsampling module and loss is introduced.

### 3.1 Overall Framework

First, we feed an image and a natural language expression into the image encoder and language encoder to generate image features and language features. Fig.2 (a) shows the overall framework. Then, the Neck fuses these two modalities into multi-modality features $F_m \in \mathbb{R}^{\frac{H}{8} \times \frac{W}{8} \times C}$, where $H$ and $W$ are the height and width of the input image and $C$ is the channel dimension. Then, the

---

[1]Code: https://github.com/toyottttttt/referring-segmentation.

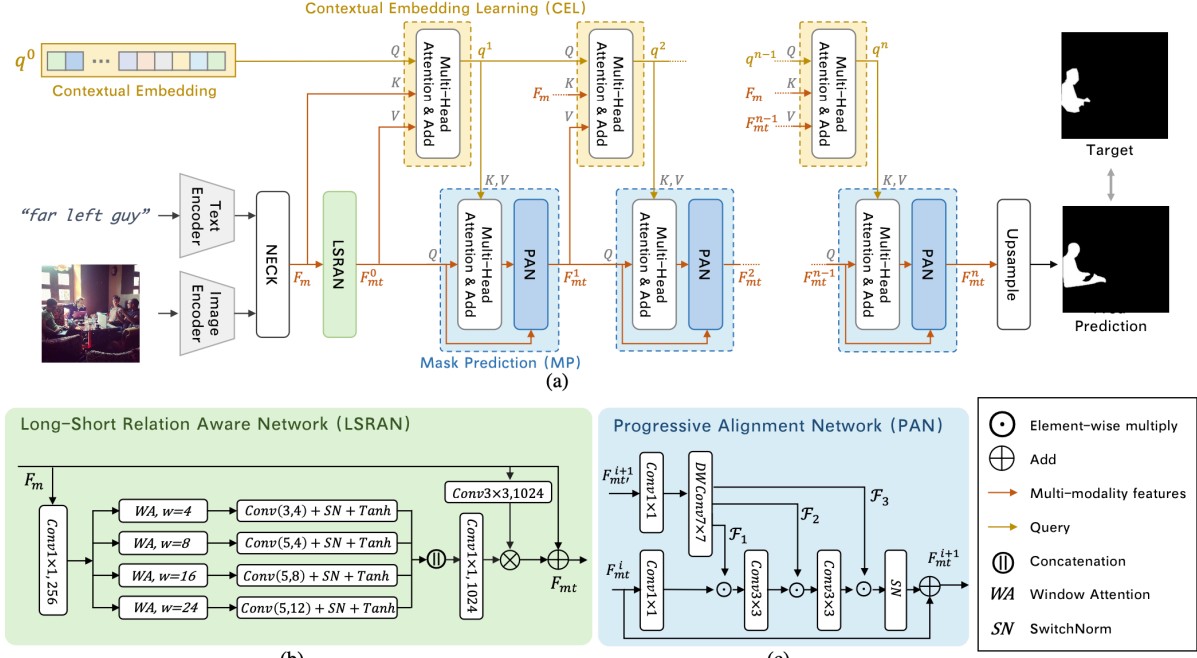

Figure 2: (a) Overall framework of our methods. It consists of a text encoder for the natural language expressions and an image encoder for the input images. Language features and image features are fused by the neck. The Long-Short Relation-Aware Network (LSRAN) captures long and short relations between objects in the image. We employ a cascade framework. $q_0$ is updated in each stage. (b) Details of LSRAN. $Conv(k, d)$ means convolutions with a kernel size of $k$ and dilated rate of $d$. $w$ is the window size for window attention. (c) Details of the Progressive Alignment Network (PAN).

spatial relations between objects in one image are estimated by using the LSRAN based on $F_m$. The output is $F_{mt} \in \mathbb{R}^{\frac{H}{8} \times \frac{W}{8} \times C}$. Unlike previous methods which utilize static-text to interface with the multi-modality output of each stage, we use Learnable Contextual Embedding $q^0 \in \mathbb{R}^{Q \times C}$, where Q is the dimensionality of contextual embedding to control the description ability. Each contextual embedding is randomly initialized by different parameters which means the information taken by different contextual embedding is not exactly same. The information carried by $q^i$ is updated to $q^{i+1}$ in accordance with $F_m$ and $F_{mt}^i$. $q^{i+1}$ serves as a K, V to update $F_{mt}^i$ into $F_{mt'}^{(i+1)}$. $F_{mt'}^{(i+1)}$ is fed into the Progressive Alignment Network (PAN) to obtain the output $F_{mt}^{(i+1)}$. There are $n$ stages in total. Finally, we upsample the refined $F_{mt}^n$ as the final output.

### 3.2 Image & Text Encoder and Neck

The referring image segmentation task has two inputs: image $I \in \mathbb{R}^{H \times W \times 3}$ and natural language expression $L \in \mathbb{R}^T$, where $T$ is the number of words in the natural language expression.

The image encoder uses the third-, fourth-, and fifth-layer feature maps of ResNet(He et al., 2016).

The maps represent $F_{v_1} \in \mathbb{R}^{\frac{H}{8} \times \frac{W}{8} \times C_1}$, $F_{v_2} \in \mathbb{R}^{\frac{H}{16} \times \frac{W}{16} \times C_2}$ and $F_{v_3} \in \mathbb{R}^{\frac{H}{32} \times \frac{W}{32} \times C_3}$, where $C_i$ denotes the channel dimension of of the 3rd-5th layers. The Transformer(Vaswani et al., 2017) is used as the language encoder to extract the global textual representation $F_s \in \mathbb{R}^{1 \times C}$ and textual representation for every word $F_t \in \mathbb{R}^{T \times C}$.

We obtain a multi-modality feature from the multi-level image features $F_{v_1}$, $F_{v_2}$, $F_{v_3}$ and global textual feature $F_s$ in a top-down manner. We reshape and repeat the global textual feature $F_s$, $\frac{H}{32} \times \frac{W}{32} \times C$ and multiply the result with the highest level image feature $F_{v_3}$ to form multi-modal features (Eq.1). Then, the lower level feature $F_{v_2}$ and $F_{v_1}$ are blended into the multi-modal feature by using Eq.2, where $i \in [1, 2]$. Here, $\sigma(\cdot)$ means batch normalization(Ioffe and Szegedy, 2015) with LeakyReLu, $Up(\cdot)$ denotes $2\times$ upsampling, $[\cdot, \cdot]$ means concatenation, while $Ws$ is a linear embedding and $W_{v_i}$ and $W_{m_i}$ are convolution layers for channel projection. The final output is $F_m = F_{m_0} \in \mathbb{R}^{\frac{H}{8} \times \frac{W}{8} \times C}$.

$$F_{m_2} = \sigma(F_s W_s) \odot \sigma(F_{v3} W_{v3}) \tag{1}$$

$$F_{m_{i-1}} = \sigma(W_{m_i}([Up(F_{m_i}), \sigma(F_{v_i} W_{v_i})])) \tag{2}$$

### 3.3 Long-Short Relation-Aware Network

There are likely to be various relations in one image. A relation between two distant objects is called a long relation, whereas a relation between two nearby objects is called a short relation. To better represent long relations and short relations, we propose the Long-Short Relation-Aware Network (LSRAN), as depicted in Fig.2 (b). Different from previous work, LSRAN aims to estimate object-wise relationships, instead of traditional pixel-wise relationship analysis (e.g., ASPP(Chen et al., 2017)), on feature maps for correctly identifying target objects. Considering that, we use a window attention(Liu et al., 2021a) to transfer information to every pixel inside window to identify semantic meaning. Then, the LSRAN employs dilated convolutions to capture the different types of relations between objects. We concatenate the four different relations and mix their channels by using a $1 \times 1$ convolution. To obtain boundary information more precisely, we feed $F_m$ into a $3 \times 3$ convolution and multiply the results with the relation features obtained. The final output is $F_{mt}$.

### 3.4 Joint Mask Contextual Embedding Learning Network

The total framework for Joint Mask and Contextual Embedding Learning Network is shown in Fig.2 (a). It is composed of two parts: contextual embedding learning and mask predicting with the Progressive Alignment Network.

#### 3.4.1 Contextual Embedding Learning

Different from the previous work(Luo et al., 2020a; CRI, 2022; Yang et al., 2021; Ding et al., 2021) which uses a static text for reasoning, we introduce a learnable embedding $q^0$ that can be updated adaptively based on the output of the current stage. Here, we use $n$ to denote the total number of stages and $i$ to represent the order of stage. The contextual embedding learning is based on multi-head attention(Vaswani et al., 2017). The $(i + 1)$-th stage Contextual Embedding is updated from the $i$-th stage Contextual Embedding, multi-modality features $F_m$, and the output of the $i$-th features $F_{mt}^i$. $F_m$ provides the initial multi-modality features and $F_{mt}^i$ provides reasoning results in the $i$-th stage. Eq.3 is the $(i+1)$-th query generation process. The target for contextual embedding learning is to hook the most expedient information based on the current reasoning output. Because initial Contextual Embedding is random randomly initialized, so it

has no useful information. Multiplication with $F_m$ leds to a reasonable correlation (attention map) between contextual embedding and original text and visual features. This attention map is applied to $F_{mt}$ to reduce distinct responses and discard unnecessary information.

$$q^{i+1} = softmax(\frac{q^i F_m^T}{\sqrt{d_k}})F_{mt}^i + q^i \quad (3)$$

#### 3.4.2 Mask Predicting

**Multi-Head Attention** After obtaining the updated $q^{i+1}$, we refresh $F_{mt}^i$ to $F_{mt}^{i+1}$. $q^{i+1}$ already contains the information required for generating $F_{mt}^{i+1}$, so we set $F_{mt}^i$ as the Q and $q^{i+1}$ as the K and V in the multi-head attention to obtain $F_{mt'}^{i+1}$, as shown in Eq.4.

$$F_{mt'}^{i+1} = softmax(\frac{F_{mt}^i (q^{i+1})^T}{\sqrt{d_k}})q^{i+1} + F_{mt}^i \quad (4)$$

**Progressive Alignment Network** Inspired by HorNet(Rao et al., 2022), we propose a progressive alignment network for a two-stage feature alignment and update. Our progressive alignment network contains three layers, as shown in Fig.2 (c). First, $F_{mt'}^{i+1}$ is projected by a $1 \times 1$ convolution into $(\frac{C}{4} + \frac{C}{2} + C)$ in channel dimension. After a $7 \times 7$ depthwise convolution, we split $F_{mt'}^{i+1}$ into three parts whose dimensions are $[\mathcal{F}_1 \in \mathbb{R}^{\frac{H}{8} \times \frac{W}{8} \times \frac{C}{4}}, \mathcal{F}_2 \in \mathbb{R}^{\frac{H}{8} \times \frac{W}{8} \times \frac{C}{2}}, \mathcal{F}_3 \in \mathbb{R}^{\frac{H}{8} \times \frac{W}{8} \times C}]$. Here, the output of the previous stage $F_{mt}^i$ is also projected by a $1 \times 1$ convolution into $\frac{C}{4}$ in channel dimension. The output is multiplied by $\mathcal{F}_1$ and the result is fed into a $3 \times 3$ convolution with $\frac{C}{2}$ in channel dimension to form a new filter for $\mathcal{F}_2$. This process is repeated three times to reduce inconsistent responses in $F_{mt'}^{i+1}$ progressively. The output produced by filtering is fed into SwitchNorm(Luo et al., 2019) and then $F_{mt}^i$ is added to form the final output $F_{mt}^{i+1}$. In the PAN, the output from the previous layer acts as filters to reduce the responses of inconsistent pixels predicted by the current layer progressively, which in turn forces the query to generate more discriminative information for the next stage.

### 3.5 Training Loss

After the multi-stage reasoning decoder, we already have reasonable results with accurate locations. We progressive upsample reasoning results into original size. The output feature map is $F \in \mathbb{R}^{H \times W \times 2}$, and the target mask is $M \in \mathbb{R}^{H \times W}$. Here, we use the cross-entropy loss to regress the final segmen-

tation mask.

# 4 Experiments

**Datasets** To verify the effectiveness of our approach, we conducted experiments on three commonly used datasets: Refcoco(Yu et al., 2016), Refcoco+(Yu et al., 2016), Refcocog(Mao et al., 2016). The Refcoco dataset targets 50,000 different objects in 19,994 images with 142,209 unique natural language expressions, which means each image contains more than one instance and each instance is described by more than one natural language expressions. Different from the Refcoco dataset, Refcoco+ has no location words in their referring expressions, which means we can locate instances by their attributes only. The length of a sentence in Refcoco+ is longer than in Refcoco. Refcoco+ consists of 141,564 expressions, 49856 objects and 19,992 images. The natural language expressions for these two datasets are generated by a two-player game(Kazemzadeh et al., 2014). Refcocog is the most challenging one among the three. Refcocog utilizes Mechanical Turk to generate natural language expressions. The dataset contains richer and longer descriptions. There are 104,560 expressions that describe 54,822 objects in 25,711 images. The average length of the referring expressions is 8.3 words. We used the UNC partition on the Refcocog dataset. The images of the three datasets were collected from MS-COCO(Lin et al., 2014).

**Implementation Details** The text and image encoder were initiated by CLIP(Radford et al., 2021b). For the image encoder, we employed ResNet-101(He et al., 2016). All of the experiments are conducted on two A100 GPUs. In what follows, $n$ represents the number of stages of the Joint Mask and Contextual Embedding Learning Network and it is set to $n$=6. Q represents the number of Contextual Embedding. Here, it is set to Q=20. The channel dimension $C$=1024 unless otherwise stated. The input images were resized to 416×416. For all three datasets, Refcoco, Refcoco+ and Refcocog, we set all sentence lengths to 30. All multi-head attentions had eight heads. The model was trained by the AdamW(Loshchilov and Hutter, 2017) optimizer for 60 epochs with a batch size of 20. The initial learning rate was 0.000006 and it was decreased by polynomial learning rate decay with power 0.9. $argmax$ along the channel dimension of the score maps was used as the prediction during inference. Following previous methods,

we train our model on its training set, choose the model's parameters with the best mIoU on the val split, and evaluate on test set. Evaluation metric details are show in Supplymentary Material.

## 4.1 Main results

Tab.1 compares the mIoUs of our method with those of other state-of-the-art methods. For a fair comparison, we report mIoU scores for **all methods** by standardized the data preprocessing method and trained previous methods using their released source code, strictly adhering to the training process outlined in their papers[2]. Our method outperformed the previous methods on almost all datasets. On the Refcoco dataset, our method surpassed the state-of-the-art LAVT(Yang et al., 2022) by 0.7% and 2.02% on val and testA splits, respectively, while all methods obtained similar mIoUs on the testB split. Moreover, it outperformed LAVT by 2.77%, 4.38%, and 0.52% on val, testA, and testB of the Refcoco+ dataset, respectively. On the most challenging dataset, Refcocog, which has significantly longer sentences, our model also achieved a noticeable improvement of 1.84% and 2.11% on the val and test splits. It is evident that our method demonstrates superior performance on more challenging datasets, particularly Refcoco+ and Refcocog, which are characterized by longer sentence lengths and more attribute descriptions.

It's worth mentioning that it outperformed CRIS(CRI, 2022) by a large margin, around 4%, on all datasets and splits. CRIS(CRI, 2022) utilizes the same pre-trained parameters as our method and it also employs the cascade framework; however, it keeps a input language features for every stage static. These results demonstrate the effectiveness of our method.

## 4.2 Ablation Study

**The Proposed Components** Tab.2 show the experiments on Refcoco testB to verify the effectiveness of our proposed model. We train our model on its training set, choose the model's parameters with the best mIoU on the val split, and evaluate on test set. Due to the space limit, we report the ablation experiments on Refcoco testB as it is considered the most difficult one, which better demonstrates the effectiveness of our proposed modules. 1 denotes our baseline whose design only contains image encoder, language encoder, neck and upsampleing

---

[2]more details can be found in supplementary material.

| Methods | Year | Refcoco | | | Refcoco+ | | | Refcocog | |
|---|---|---|---|---|---|---|---|---|---|
| | | val | testA | testB | val | testA | testB | val | test |
| RMI(Liu et al., 2017) | ICCV'17 | 41.98 | 43.46 | 42.56 | 24.98 | 26.89 | 24.91 | 31.78 | 31.70 |
| RRN(Li et al., 2018) | CVPR'18 | 54.33 | 56.12 | 52.78 | 37.22 | 40.24 | 34.12 | 41.22 | 41.90 |
| CMSA(Ye et al., 2019) | CVPR'19 | 59.00 | 62.22 | 55.99 | 40.41 | 45.19 | 34.60 | 40.07 | 40.27 |
| BRINet(Hu et al., 2020) | CVPR'20 | 62.74 | 65.23 | 60.80 | 49.32 | 54.20 | 43.20 | 47.55 | 47.74 |
| CMPC(Huang et al., 2020) | CVPR'20 | 63.82 | 66.78 | 61.49 | 52.08 | 56.83 | 45.84 | 52.66 | 53.27 |
| LSCM(Hui et al., 2020) | ECCV'20 | 63.46 | 65.88 | 61.52 | 52.01 | 56.51 | 45.24 | 50.87 | 51.15 |
| MCN(Luo et al., 2020b) | CVPR'20 | 62.91 | 64.88 | 60.22 | 52.19 | 56.43 | 46.99 | 48.64 | 49.22 |
| CGAN(Luo et al., 2020a) | MM'20 | 64.86 | 68.04 | 62.07 | 51.03 | 55.51 | 44.06 | 51.01 | 51.69 |
| LTS(Jing et al., 2021) | CVPR'21 | 65.43 | 67.76 | 63.08 | 54.21 | 58.32 | 48.02 | 54.40 | 54.25 |
| EFN(Feng et al., 2021) | CVPR'21 | 65.02 | 65.98 | 63.28 | 53.18 | 56.45 | 47.31 | 55.01 | 54.85 |
| VLT(Ding et al., 2021) | ICCV'21 | 65.65 | 68.29 | 62.73 | 55.50 | 59.20 | 49.36 | 52.99 | 56.65 |
| CRIS(CRI, 2022) | CVPR'22 | 70.28 | 73.39 | 65.41 | 62.92 | 67.75 | 53.24 | 60.72 | 61.04 |
| LAVT(Yang et al., 2022) | CVPR'22 | 73.70 | 75.67 | **70.44** | 64.22 | 68.31 | 56.84 | 62.24 | 62.88 |
| Ours | - | **74.40** | **77.69** | 70.43 | **66.99** | **72.69** | **57.34** | **64.08** | **64.99** |

Table 1: Comparison with state-of-the-art methods on three widely used datasets. We trained all methods on their released code with 416×416 input resolution and used mIoU as the evaluation metric.

| # | Methods | mIoU | Pr@50 | Pr@90 |
|---|---|---|---|---|
| 1 | Baseline | 62.57 | 70.11 | 23.73 |
| 2 | +LSRAN | 68.03 | 77.53 | 27.36 |
| 3 | +Cascade | 69.23 | 78.08 | 29.54 |
| 4 | Ours(w/o PAN) | 69.46 | 78.86 | 30.56 |
| 5 | Ours(w/ PAN) | **70.43** | **79.88** | **31.54** |
| 6 | w/o adaptive LCE | 69.11 | 77.60 | 29.22 |

Table 2: Ablation study on Refcoco testB.

module. In order to enhance the baseline performance, we introduce LSRAN, a novel approach that incorporates the capability to capture relations between objects within a single image. Our proposed method not only achieves a significant improvement of approximately 5% in mIoU, but also enhances the target rate (Pr@50) and mask quality (Pr@90). We further investigate the impact of integrating the Cascade framework with Learnable Contextual Embedding, and present the results in 3. The Cascade framework with Learnable Contextual Embedding leads to a significant improvement of approximately 7% in mIoU which demonstrate the effectiveness of combining the Cascade framework and Learnable Contextual Embedding in improving the quality of reasoning outcomes.

In 4, we take a step further by combining LSRAN with the Cascade framework. In this hybrid approach, the outputs from the neck component solely provide image and language information, while LSRAN introduces additional relation information. By incorporating LSRAN into the Cascade framework, we slight improve mIoU, targeting rate and mask quality. To address the issue of error propagation from early stages, we introduce the Progressive Alignment Network (PAN) at the end of each stage. In comparison to the results presented in Experiment 4, Experiment 5 showcases the effectiveness of PAN in rectifying error propagation. Specifically, PAN demonstrates its ability to improve the mean mIoU by approximately 1%, increase the targeting rate by 1.02%, and enhance mask quality. These findings highlight the important role of PAN in mitigating the impact of error propagation and further improving the performance of the overall system.

In Experiment 6, we investigate the effectiveness of the feedback mechanism by excluding the update of Learnable Contextual Embedding (LCE) based on each stage's output. Instead, we input the previous stage's fused features, denoted as $F_m$ and $F_{mt}^0$, into each subsequent stage. The purpose of this evaluation is to assess the impact of feedback on the overall performance. Surprisingly, the results reveal a decrease in mIoU by approximately 1.32% compared to previous experiments. This indicates that updating the Learnable Contextual Embedding based on each stage's output plays a crucial role in refining the reasoning process and improving the accuracy of the segmentation task.

## 5 Visualization

Fig.3 shows some qualitative results, comparing our method and previous state-of-the-art methods.

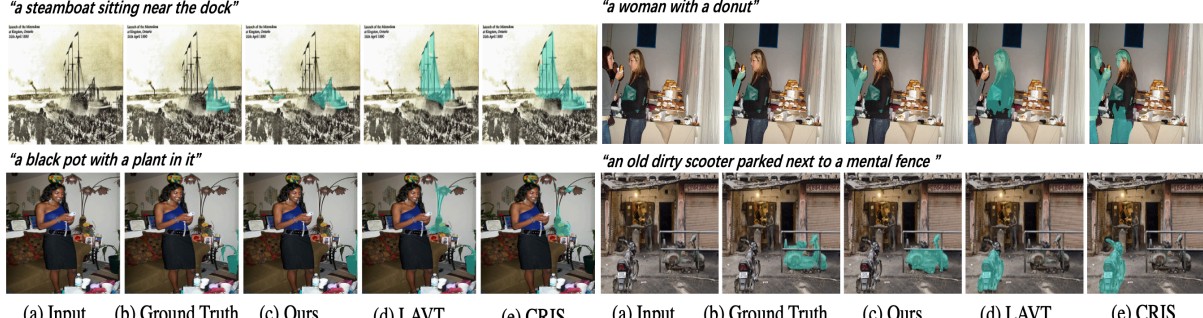

(a) Input    (b) Ground Truth    (c) Ours    (d) LAVT    (e) CRIS      (a) Input    (b) Ground Truth    (c) Ours    (d) LAVT    (e) CRIS

Figure 3: Qualitative comparison of our work and previous work: LAVT and CRIS.

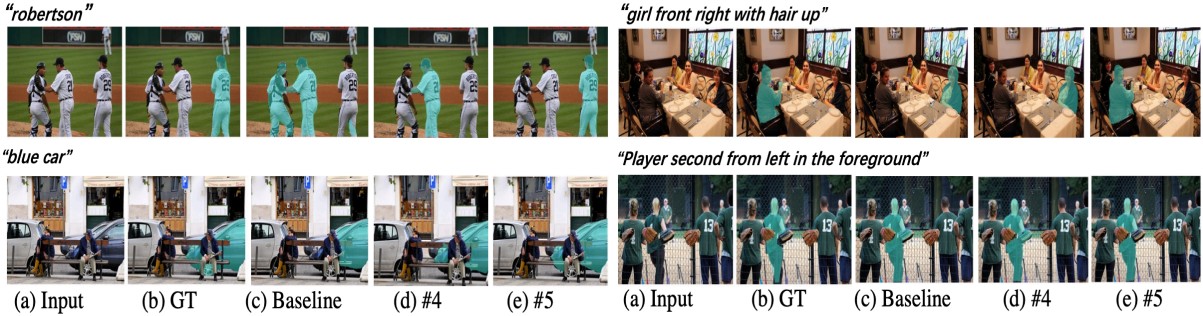

(a) Input    (b) GT    (c) Baseline    (d) #4    (e) #5      (a) Input    (b) GT    (c) Baseline    (d) #4    (e) #5

Figure 4: Qualitative comparison of ablation study. #4 and #5 indicates the number in Table 2.

| Model | Split | mIoU | Pr@50 | Pr@90 |
|-------|-------|------|-------|-------|
| LAVT | testA | 68.31 | 78.31 | 26.77 |
| Ours | | **72.69**(+4.38) | **83.74**(+5.43) | **29.71**(+2.94) |
| LAVT | testB | 56.84 | 63.24 | **24.34** |
| Ours | | **57.34**(+0.50) | **64.31**(+1.07) | **24.34**(+0.00) |

Table 3: Results for refcoco+, testA and testB.

While LAVT and CRIS encountered difficulties and made mistakes in these four challenging cases, our method demonstrated robust performance and successfully tackled them.

Figure 4 presents qualitative results obtained from the ablation study. Starting from the baseline and progressing to experiment #4, the Cascade framework with Learnable Contextual Embedding is introduced. This addition results in improved performance, as observed in the case of the "blue car" where the Cascade framework with Learnable Contextual Embedding predicts a more compact mask compared to the baseline. Experiment #5 introduces the Progressive Alignment Network (PAN). Notably, in the cases of "robertson" and "girl front right with hair up", PAN demonstrates its ability to rectify incorrect predictions during the reasoning process. This showcases the effectiveness of PAN in refining and improving the accuracy of the segmentation outputs.

## 6 Limitations

As shown in Tab.3, our methods produce a relative improvement in the mIoU metric of 4.38% in testA and 0.50% in testB. Moreover, in testA, there is 4.38% improvement in Pr@50, while Pr@90 only improves by 2.94%. The improvement in Pr@50 in testB is 1.07%, while Pr@90 shows no improvement. By analyzing testA and testB, we find the target in testA is the human instances whose size are larger while the target in testB is mainly objects whose size are smaller. This indicates that our methods have more ability to identify larger target objects, but for the smaller objects, the improvement is limited. And our method is not good at producing high-quality output mask. Compared with LAVT, our cascade stages only uses $52 \times 52$ feature maps for reasoning, while LAVT utilizes a clear-sighted early fusion method, in which the query is fused with a larger feature map with a $104 \times 104$ size in the first stage. The larger feature map is good for finding smaller objects and keeping the boundary. However, because the number of layers in image encoding backbone designed for other task is fixed, it is difficult for early fusion methods to define their own reasoning stages, which limits their performance. For example, Swin-Transformer(Liu et al., 2021b) has four layers, so LAVT performs image-text alignment four times. By comparison, we adopt the late fusion cascade

framework which is more flexible in defining the number of reasoning stages fit for this task.

## 7 Conclusion and Future Work

We proposed a Joint Mask and Contextual Embedding Learning Network that dynamically updates the Contextual Embedding on the basis of the reasoning output and a long-short relation-aware network to learn objects relations. Experimental results demonstrate the effectiveness of our methods. In the future, we want to handle the limitation mentioned in Sec.6 and improve the targeting rate for small objects and recover boundary information better.

## 8 Acknowledge

Ziling Huang was supported by Japanese Government(MEXT) Scholarship and this work was supported by JSPS KAKENHI JP22H03620 and JP22H05015.

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
