# OpenReview forum: "Referring Image Segmentation via Joint Mask Contextual Embedding Learning and Progressive Alignment Network"
_EMNLP/2023/Conference — EMNLP 2023 Main_

### Official Review · Reviewer_hADV · 2023-08-04

**Soundness:** 4

**Excitement:**

3: Ambivalent: It has merits (e.g., it reports state-of-the-art results, the idea is nice), but there are key weaknesses (e.g., it describes incremental work), and it can significantly benefit from another round of revision. However, I won't object to accepting it if my co-reviewers champion it.

**Missing References:**

Here are some suggestions to improve spelling, grammar, writing style, and presentation of the paper:
Line 78: Rephrase "its defects also obvious" to "its limitations are also apparent" for better clarity.
Line 61: Add "the" before "early fusion framework".
Line 139: Change "Endow the network" to "Endows the network" to fix grammar error.
Line 316: Simplify phrase "obtained before" to just "obtained".

**Paper Topic And Main Contributions:**

This paper investigates the task of referring image segmentation, aiming to predict the pixel-level occlusion of the objects mentioned in the image based on natural language descriptions. The main contributions of this paper are:
Proposing a Joint Multi-stage Contextual Embedding Learning Network (JMCELN), which can dynamically store and utilize multi-stage inference information through learnable context embedding.
Introducing a Progressive Alignment Network (PAN), which can correct the inconsistencies in predictions at different stages, reducing error propagation.
Conducting experiments on three referring segmentation datasets, the results outperform multiple state-of-the-art methods, demonstrating the effectiveness of the proposed approach.
Proposing a Long-Short Relation Awareness Network (LSRAN), which can better capture the relationships between objects within the image.
This paper addresses the issues of over-reliance on static language expressions and error propagation inherent in current multi-stage methods. By implementing learnable context embedding and the Progressive Alignment Network, it achieves more accurate and robust multi-stage inferences, representing high-quality innovative work in this field.

**Questions For The Authors:**

Question A: While the performance gains on the Refcoco testB dataset are noteworthy, could you provide further results from other datasets to help validate the method's effectiveness and generalizability?
Question B: In terms of transformer-based segmentation methods, particularly the recently popular Meta-SAM, could you outline the advantages of the proposed approach? It would be beneficial to highlight the unique strengths of your method.
Question C: Visualization of instances where the method does not perform as expected might offer a valuable perspective on the current limitations. It could also help in identifying potential areas for improvement.
Question D: Some parts of the paper could benefit from further refinement to enhance the readability and logical flow. For instance, it might be helpful to consolidate some sections in the methods part.
I hope these queries will aid the authors in enhancing the strength of the paper.

**Reasons To Accept:**

A Joint Multi-stage Contextual Embedding Learning Network, which can dynamically update context embeddings and better perform multi-stage inference, offering an effective framework for referring image segmentation tasks.
A Progressive Alignment Network has been designed to effectively reduce error propagation across different stages and improve the consistency of prediction results. This design concept can also be generalized to other visual tasks that require multi-stage inferences.
Comprehensive validation was conducted on multiple referring image segmentation datasets, with all results outperforming the current state-of-the-art methods, proving the effectiveness of the proposed method.
A Long-Short Relation Awareness Network was introduced to better represent the relationships between objects within images. This component design can also be generalized to other tasks that require object relationship modeling.
The results of the paper can provide reference on how to better integrate visual and language information and carry out multi-stage inference in other image understanding tasks.
The code and model of the paper will be publicly released, providing other researchers with powerful and effective pre-trained models to advance the development of this field.
Overall, this paper has made valuable explorations in advancing multi-stage inference, error propagation correction, and the combination of image understanding and natural language, and has the potential to be published at top-tier conferences.

**Reasons To Reject:**

The proposed Joint Mask Contextual Embedding Learning Network (JMCELN) expands upon previous cascade frameworks. It would be beneficial to further elucidate the unique aspects that separate JMCELN from these existing frameworks.
While the Learnable Contextual Embedding (LCE) and Progressive Alignment Network (PAN) demonstrate improved performance, more extensive analysis and additional ablation studies could provide further insight into their effectiveness and indispensability.
The performance improvements on some datasets are notable but could be more consistent across all tested datasets. A deeper exploration of the method's generalizability could strengthen the paper.
Certain areas of the paper could benefit from further refinement to enhance readability and logical flow.
In summary, this paper tackles an important issue and contributes valuable advancements. To further strengthen the paper, consider focusing on demonstrating the novelty of the method, ensuring completeness in evaluation, and enhancing the clarity and organization of the writing. I hope these suggestions will assist the authors in improving the paper.

**Reproducibility:**

4: Could mostly reproduce the results, but there may be some variation because of sample variance or minor variations in their interpretation of the protocol or method.

**Reviewer Confidence:**

3: Pretty sure, but there's a chance I missed something. Although I have a good feel for this area in general, I did not carefully check the paper's details, e.g., the math, experimental design, or novelty.

---

> ### Author Rebuttal · Authors · 2023-08-24
>
> We greatly appreciate your insightful feedback, which has played a crucial role in improving the quality of our paper.
>
> (1) Ablation Study for JMCELN:  When remove JMCELN from original cascade network, the model become original static-text stage reasoning network. We obtained 68.77 mIoU in Refcoco testB, and 76.32 mIoU in Refcoco testA dataset. Compared with results in our paper, our error correction mechanism can largely improve the final results of cascade network.
>
> (2) Question A:
> ***
> |# | Methods              | mIoU | Pr@50| Pr@90 |
> | --- | --- | --- | --- | --- |
> | 1| Baseline              | 71.68 | 81.46  | 26.29   |
> | 2| +LSRAN              | 75.37 | 87.29  | 29.10   |
> | 3| +Cascade            | 76.85 | 88.70  | 30.35.  |
> | 4| Ours(w/o PAN)    | 77.21 | 88.99  |  31.15  |
> | 5| Ours(w/ PAN)      | 77.69 | 89.27  |  31.31  |
> | 6| w/o adaptive LCE | 77.04| 89.04  | 29.57   |
> ***
>
> Due to time limitation, the table provided represents the ablation study conducted on the refcoco testA dataset, which is known as the easiest split within the refcoco dataset. From table, it is obvious, the proposed LSRAN, LCE and PAN can help model improve mIoU, targeting rate(Pr@50) and boundary precision(Pr@90), which help validate the method's effectiveness and generalizability. In camera ready version, we will provide more ablation results for refcoco+ and refcocog dataset.
>
> (3) Question B: Strengths compared with Meta-SAM
>     (a) Our main advantage over Meta-SAM lies in the fact that our model can understand the meaning of text inputs better and segment the object text referred to. In the context of Meta-SAM, although they offer prompts in the form of points, bounding boxes, and text, they have not yet released the model specifically designed for text prompts. Consequently, the current Meta-SAM framework remains incapable of effectively performing the task of referring segmentation. To assess the efficacy of their text prompts, we extracted BERT text features as prompts and input them into the Meta-SAM framework. We utilized the refcoco dataset for fine-tuning and implemented various training strategies. However, even after these rigorous efforts, the highest achieved mIoU on the validation set only approximately 58%, a considerable gap compared to our model's performance at 74.40%.
>     (b) Although Meta-SAM achieved notable segmentation results, much of its performance stems from the availability of extensive training data and abundant computing resources. They employed 128 GPUs for training the Meta-SAM model, a resource-intensive endeavor that might be financially impractical for ordinary research institutes. In contrast, our model relies solely on publicly available datasets such as refcoco, refcoco+, and refcocog for training. Remarkably, our entire training process is accomplished within a day on 4 GPUs.
>
> (4) Question C: Visualization of Failure Cases
> We has examined failure cases, revealing insights into specific challenges. Two primary categories emerge: (a) Small Object: We already confirmed that our method can properly handle the cases when the target objects are small, but final segmentation map can not find its precise boundary because input feature maps are too small.  (b) Incorrect Text: In cases where text contains errors or abbreviations, such as "appl" instead of "apple," our model encounters difficulties in establishing accurate connections. To offer a well-rounded assessment, we visually present these failure scenarios in our paper. These insights deepen our understanding of our approach's strengths and limitations, guiding future advancements.
>
> (5) Question D:
> We greatly appreciate your valuable feedback regarding the refinement of certain sections within our paper. Your insights are instrumental in guiding us toward improving the overall readability and logical coherence. We are committed to carefully examining these sections and making necessary adjustments to ensure paper's readability.

---

### Official Review · Reviewer_mgwW · 2023-08-05

**Soundness:** 3

**Excitement:**

3: Ambivalent: It has merits (e.g., it reports state-of-the-art results, the idea is nice), but there are key weaknesses (e.g., it describes incremental work), and it can significantly benefit from another round of revision. However, I won't object to accepting it if my co-reviewers champion it.

**Paper Topic And Main Contributions:**

The paper proposes a joint mask contextual embedding learning network, JMCELN, to solve the challenges in both maintaining a strong focus on the most relevant information at specific stages and  rectifying errors propagated from early stages. JMCELN proposes to use a learnable contextual embedding module to enable adaptive capturing pertinent information, and a progressive alignment network to reduce inconsistencies between stages. Experiments show that the proposed methods outperform baseline methods on three benchmarks.

**Reasons To Accept:**

1. The paper is well-written, making it easy to follow the proposed method and its contributions.
2. The drawn diagram visually presents the proposed approach, enhancing understanding.
3. JMCELN demonstrates the capability to improve the segmentation of large objects effectively.

**Reasons To Reject:**

1. The main concern is that JMCELN appears to be restricted to improving the segmentation of large objects, rendering it unsuitable for generating scenarios involving smaller objects.
2. The paper compares JMCELN with works up to 2022, disregarding newer approaches like PolyFormer [1], which achieve superior performance in early 2023.

**Reproducibility:**

4: Could mostly reproduce the results, but there may be some variation because of sample variance or minor variations in their interpretation of the protocol or method.

**Reviewer Confidence:**

3: Pretty sure, but there's a chance I missed something. Although I have a good feel for this area in general, I did not carefully check the paper's details, e.g., the math, experimental design, or novelty.

---

> ### Author Rebuttal · Authors · 2023-08-29
>
> Thank you for your valuable feedback on our paper.
>
> (1) Enhancement for Smaller Objects:
> Our proposed framework can not only handle large objects, but also effectively in finding small object. After carefully check, the Table 3 in our paper, in Refcoco testA(first row),  around 2.31% targeting rate(Pr@50) improvements is contributed by small objects while large   objects contribute 3.12% improvement in targeting rate(Pr@50). While in Refcoco testB(second row),  the small objects contribute 0.43% improvements in targeting rate(Pr@50), and large objects contribute 0.64% improvements in targeting rate(Pr@50). These results mean our proposed model also improve the targeting rate for small objects. While in mask quality(Pr@90), small objects contribute 1.2% improvement in Refcoco testA, and around 1.7% mask quality(Pr@90) improvement is contributed by larger objects. (The larger objects and small objects are defined by area provided by mscoco dataset, area<481 is small object while area>=481 is larger object. )
>
> (2) Acknowledging Recent SOTA Development in Early 2023:
> Thank you for your suggestions. We also notice Polyformer on arxiv in Feb. 2023.  However, it is unfair to compare our work with Polyformer and two work, SeqTR[1] published in ECCV2020 and RefTr[2] published in NIPS2021, because of two reasons: (a) all these work use large-scale referring grounding datasets Visual Genome and Flickr30k-entities to pretrain their model. (b) all these methods are designed for multitask. As shown in SeqTR[1] and RefTr[2],  the pretraining can bring approximately 5% improvement in mIoU for refcoco and an impressive 10% improvement in mIoU for refcoco+ and refcocog. Moreover, in multitask methods, training referring grounding and referring segmentation jointly would significantly impact the performance of referring segmentation, as evidenced in Table 5 of the PolyFormer and Table 4 in RefTr[2]. Because of these reasons, most of previous work[3][4] in referring segmentation do not compare with these work.
>
> It's crucial to highlight that our work only focuses on referring segmentation and does not incorporate the large-scale referring grounding dataset for pretraining. Therefore, a direct comparison between our work and PolyFormer would be unfair.
>
> We greatly value your engagement and the insights you provide. Your comments substantially enrich the depth and quality of our research, propelling us to continuously refine and enhance our work.
>
> [1] SeqTR: A Simple Yet Universal Network for Visual Grounding. ECCV2020
>
> [2] A one-step approach to multi-task visual grounding. NIPS2021
>
> [3] CRIS: CLIP-Driven Referring Image Segmentation. CVPR2022
>
> [4] LAVT: Language-Aware Vision Transformer for Referring Image Segmentation.CVPR2022

---

### Official Review · Reviewer_i668 · 2023-08-07

**Soundness:** 4

**Excitement:**

4: Strong: This paper deepens the understanding of some phenomenon or lowers the barriers to an existing research direction.

**Paper Topic And Main Contributions:**

This paper presents a novel approach to address the problem of stage reasoning in Mask Predicting tasks, particularly in the context of referring expression comprehension. The paper introduces an innovative solution to the stage reasoning problem in Mask Predicting tasks by leveraging Learnable Contextual Embeddings and Progressive Alignment Network. The proposed approach achieves state-of-the-art results on challenging referring expression comprehension datasets, providing valuable insights into the field of visual reasoning and language understanding.

**Reasons To Accept:**

1) The introduction of JMCELN, which replaces static-text with Learnable Contextual Embeddings, is a significant contribution. By incorporating Learnable Contextual Embeddings, the model gains the ability to capture pertinent information at every stage of reasoning. This approach enhances the network's capacity to understand the context and generate more accurate mask predictions.
2) The proposal of the Progressive Alignment Network is another noteworthy contribution. The PAN plays a crucial role in improving the accuracy of mask predictions by addressing inconsistencies between predictions from different stages. This iterative alignment mechanism reduces the response of the later output and encourages the generation of more discriminative information, resulting in more precise masks.
3) The paper's methods, JMCELN, and PAN, achieve state-of-the-art performance on three widely used datasets in referring expression comprehension tasks: Refcoco, Refcoco+, and Refcocog. The demonstration of superior performance, especially on intricate datasets like Refcoco+ and Refcocog, highlights the effectiveness of the proposed approach.
4) The paper reinforces the validity of its proposed components through extensive ablation studies. By conducting these studies, the authors demonstrate the importance of each proposed component and its positive impact on performance.

**Reasons To Reject:**

1) While the core ideas of the proposed modules, Contextual Embeddings, and Progressive Alignment, are cited as widely referenced in related fields, the paper lacks a clear and thorough clarification of how these concepts differ from existing approaches in the context of the specific problem addressed in this paper. It would be beneficial to provide a more detailed comparison with related works to highlight the novelty of the proposed techniques.
2) The paper could benefit from a more explicit and concise description of the challenges faced in the stage reasoning problem and the precise motivations behind the design of JMCELN and PAN.
3) The visualizations presented in Figures 3 and 4 illustrate the effectiveness of the proposed method. However, the paper could be improved by providing additional examples where the proposed approach might fail or show limitations. This would allow readers to gain a better understanding of the proposed method's capability boundaries and its performance in various scenarios.

**Reproducibility:**

4: Could mostly reproduce the results, but there may be some variation because of sample variance or minor variations in their interpretation of the protocol or method.

**Reviewer Confidence:**

3: Pretty sure, but there's a chance I missed something. Although I have a good feel for this area in general, I did not carefully check the paper's details, e.g., the math, experimental design, or novelty.

---

> ### Author Rebuttal · Authors · 2023-08-29
>
> We deeply value your feedback and its contribution to enhancing the clarity of our paper.
>
> (1) Highlight the difference of the proposed techniques:
> (a) Learnable Contextual Embeddings are inspired by the use of random/zero initial queries in K-net and Segformer. However, it differs from queries in K-net and Segformer in two aspects: i) In K-Net and Segformer, the number of queries represents the number of segmentation maps in the final outputs. In contrast, the size of our Learnable Contextual Embeddings represents the capacity to carry information for reasoning.  ii) Our Learnable Contextual Embeddings can be updated based on the output of the current layer, whereas queries in K-Net and Segformer cannot be updated adaptively; they only take in information layer by layer. (b) The Progressive Alignment Network draws inspiration from the concept of progressive refinement as seen in HorNet. While HorNet operates with a single input, our Progressive Alignment Network is newly designed for utilizing two inputs from different stages, facilitating a progressive comparison of their differences. By integrating these two components, we introduce a new error correction mechanism for cascade network which is not appear in any previous work.
>
> (2) Description of Challenges and Motivations:
> Thank you for your valuable suggestions. We will incorporate the following in our introduction:  In summary, cascade network presents two key challenges: the repetition of static text accumulates noise throughout reasoning and misunderstanding reasoning at earlier stages propagate, causing inaccurate predictions in subsequent stages. To avoid introduce too much noise, we introduce Learnable Contextual Embedding. Learnable Contextual Embedding aims to provide each stage dynamic and the most pertinent information, while PAN combats inconsistencies between stages to avoid error propagation. Learnable Contextual Embedding and Progressive Alignment Network ensure the successful operation of our error correction mechanism within the cascade network.
>
> (3) Inclusion of Limitations and Failure Cases:
> We has examined failure cases, revealing insights into specific challenges. Two primary categories emerge: (a) Small Object: We already confirmed that our method can properly handle the cases when the target objects are small, but final segmentation map can not find its precise boundary because input feature maps are too small.  (b) Incorrect Text: In cases where text contains errors or abbreviations, such as "appl" instead of "apple," our model encounters difficulties in establishing accurate connections. To offer a well-rounded assessment, we visually present these failure scenarios in our paper. These insights deepen our understanding of our approach's strengths and limitations, guiding future advancements.
>
> We sincerely appreciate your thoughtful critique and the opportunity to enhance the clarity, thoroughness, and robustness of our paper. Your insights are invaluable in driving us to refine and elevate the quality of our research.

---

### Meta-Review · Area_Chair_oGtw · 2023-09-15

**Recommendation:** 5

**Metareview:**

Three reviewers provided feedback for this paper and their reviews were in consensus. Overall, reviewers found the work to be novel and significant, particularly the new network architecture. They appreciated the results, i.e. the high scores on multiple benchmarks, as well as the ablation analysis provided by the authors. The main complaints included the lack of extensive qualitative examples and a clear comparison/clarification to recent related work. There was also a concern that the method does not work well for small objects, but I think this isnt a strong concern since it is unreasonable to expect a single paper to address all concerns within a problem. Overall, post rebuttal, reviewers continue to be happy with this paper and fairly excited about it. I agree with this feedback.

---

### Decision · Program_Chairs · 2023-10-07

**Decision:**

Accept-Main

**Comment:**

Three reviewers provided feedback for this paper and their reviews were in consensus. Overall, reviewers found the work to be novel and significant, particularly the new network architecture. They appreciated the results, i.e. the high scores on multiple benchmarks, as well as the ablation analysis provided by the authors. The main complaints included the lack of extensive qualitative examples and a clear comparison/clarification to recent related work. There was also a concern that the method does not work well for small objects, but I think this isnt a strong concern since it is unreasonable to expect a single paper to address all concerns within a problem. Overall, post rebuttal, reviewers continue to be happy with this paper and fairly excited about it. I agree with this feedback.